# Reciprocal Homer1a and Homer2 Isoform Expression Is a Key Mechanism for Muscle *Soleus* Atrophy in Spaceflown Mice

**DOI:** 10.3390/ijms23010075

**Published:** 2021-12-22

**Authors:** Dieter Blottner, Gabor Trautmann, Sandra Furlan, Guido Gambara, Katharina Block, Martina Gutsmann, Lian-Wen Sun, Paul F. Worley, Luisa Gorza, Martina Scano, Paola Lorenzon, Imre Vida, Pompeo Volpe, Michele Salanova

**Affiliations:** 1Charité–Universitätsmedizin Berlin, Corporate Member of Freie Universität Berlin, Humboldt-Universität zu Berlin and Berlin Institute of Health, Institute of Integrative Neuroanatomy, 10117 Berlin, Germany; dieter.blottner@charite.de (D.B.); gabor.trautmann@charite.de (G.T.); martina.gutsmann@charite.de (M.G.); imre.vida@charite.de (I.V.); 2Neuromuscular System and Neuromuscular Signaling, Center of Space Medicine Berlin, 10115 Berlin, Germany; guidogambara@gmail.com (G.G.); katharina.block@charite.de (K.B.); 3C.N.R. Institute of Neuroscience, 35121 Padova, Italy; sfurlan@mail.bio.unipd.it; 4Beijing Advanced Innovation Center for Biomedical Engineering, School of Biological Science and Medical Engineering, Beihang University, Beijing 100083, China; sunlw@buaa.edu.cn; 5Department of Neuroscience, Johns Hopkins University, Baltimore, MD 21205, USA; pworley1@jhmi.edu; 6Department of Biomedical Sciences, University of Padova, 35131 Padova, Italy; luisa.gorza@unipd.it (L.G.); martina.scano@phd.unipd.it (M.S.); pompeo.volpe@unipd.it (P.V.); 7Department of Life Sciences, University of Trieste, 34127 Trieste, Italy; plorenzon@units.it; 8B.R.A.I.N., Centre for Neuroscience, 34127 Trieste, Italy

**Keywords:** Homer isoform switch, NMJ adaptation, microgravity, hindlimb unloading, muscle atrophy

## Abstract

The molecular mechanisms of skeletal muscle atrophy under extended periods of either disuse or microgravity are not yet fully understood. The transition of Homer isoforms may play a key role during neuromuscular junction (NMJ) imbalance/plasticity in space. Here, we investigated the expression pattern of Homer short and long isoforms by gene array, qPCR, biochemistry, and laser confocal microscopy in skeletal muscles from male C57Bl/N6 mice (*n* = 5) housed for 30 days in space (Bion-flight = BF) compared to muscles from Bion biosatellite on the ground-housed animals (Bion ground = BG) and from standard cage housed animals (Flight control = FC). A comparison study was carried out with muscles of rats subjected to hindlimb unloading (HU). Gene array and qPCR results showed an increase in Homer1a transcripts, the short dominant negative isoform, in *soleus* (*SOL*) muscle after 30 days in microgravity, whereas it was only transiently increased after four days of HU. Conversely, Homer2 long-form was downregulated in *SOL* muscle in both models. Homer immunofluorescence intensity analysis at the NMJ of BF and HU animals showed comparable outcomes in *SOL* but not in the *extensor digitorum longus* (*EDL*) muscle. Reduced Homer crosslinking at the NMJ consequent to increased Homer1a and/or reduced Homer2 may contribute to muscle-type specific atrophy resulting from microgravity and HU disuse suggesting mutual mechanisms.

## 1. Introduction

Prolonged periods of muscle disuse in different musculoskeletal disorders and myopathies, as well as aging and gravitational unloading during long-term spaceflight missions, result in significant alterations in neuromuscular junction (NMJ) structure and function, which are crucial for muscle mass control and fine motor performance [1,2,3,4,5,6]. Because of a progressive increase in the duration of spaceflight missions, prevention of NMJ deconditioning and, thus, skeletal muscle functional impairments in space, imposes great limits on the Human Space Exploration Program and represents a real mission challenge. Therefore, a deeper understanding of the molecular and cellular mechanisms leading to the NMJ imbalance in space and its prevention by adequate inflight countermeasure prescription are of paramount importance for future mission success.

Despite the identification of several molecular players at the NMJ and their specific signaling pathways and cross-talk, the molecular mechanisms whereby neurophysiological nerve impulses are translated into skeletal muscle transcriptional and mechanical responses during NMJ adaptation and plasticity in different loading, unloading, and microgravity (µg) environmental conditions are not yet fully understood. To further investigate such mechanisms, in the last two decades, the U.S. National Aeronautics and Space Administration (NASA) and the European Space Agency (ESA) organized several “Space Animal Studies” with mice and rats housed in space-qualified automated life support systems upgraded with small animal modules [7,8,9].

Previous studies, using mice and rats in space as experimental models, proposed that early events of spaceflight adaptation at skeletal muscle level are represented by presynaptic structural and functional changes. Baranski et al. [10,11] reported a decreased number of presynaptic neurotransmitter vesicles in the deep calf *SOL* in 21-days space-flown rats. Moreover, Pozdniakov et al. [12] and Riley et al. [13] reported reduced nerve terminal sprouting and abandoned postsynaptic grooves after 13 and 12.5 days of microgravity exposure, respectively. Findings were confirmed by D’Amelio and Daunton [14]. Interestingly, Deschenes et al. [15] reported on the increased NMJ structural remodeling after 10 days of microgravity unloading in postural muscles of mice. All these studies, however, were carried out at the structural and ultrastructural level, mostly focusing on the nerve presynaptic side. Studies addressing molecular changes at the NMJ postsynaptic microdomain during prolonged muscle unloading or chronic exposure to microgravity are still missing.

We previously reported that components of the Homer protein family are localized at the NMJ postsynaptic microdomain [16], whose expression is regulated by muscle and nerve activity [16,17]. At central neuronal synapses, in addition to their ability to bind metabotropic glutamate receptors (mGluRs) [18], Homer proteins were shown to interact with intracellular Ca^2+^ release channels, such as the ryanodine receptor 1 (RyR1) [19,20] and the Inositol Trisphosphate Receptors type 1 (IP_3_R1) [21,22], suggesting that Homer could directly link cellular membrane-associated signaling proteins with endoplasmic reticulum domains. Conversely, after the induction of the short Homer1a, the immediately early gene (IEG) isoform that lacks a coiled-coil (CC) domain disrupted constitutively-expressed Homer long isoforms playing a key role as a dominant negative regulator [23,24]. Therefore, due to the unique ability of long Homer monomers to oligomerize to form homo- or hetero-dimers and/or multimers [16,25] and to interact simultaneously with different molecular partners [26], the tight expression and subcellular localization regulation of Homer specific protein isoforms at the NMJ postsynaptic microdomain could play an eminent role as crosslinking proteins able to either promote or turn off protein signal transduction and cross-talk within the same or different signaling pathways [16]. Interestingly, the unique C-terminal sequence of Homer1a is well conserved between mouse, rat, and human species, suggesting an evolutionarily conserved function [27].

The aim of the present study was to further evaluate the molecular features of skeletal muscle NMJ adaptation and plasticity in spaceflight microgravity vs. 1 g (Earth gravitation), focusing on Homer protein family regulation and expression at the NMJ.

In 2013, the Russian biomedical research program in space flew adult male C57BL/N6 mice onboard the Bion-M1 biosatellite with the aim to evaluate the effect of 30 days of microgravity exposure on different bio-parameters in vivo and in vitro [28]. Given the opportunity to participate in the Bion-M1 tissue-sharing program (2012–2013), we investigated the expression and subcellular localization of Homer isoforms at the NMJ of both postural (*soleus*, *SOL*) and non-postural (extensor digitorum longus, *EDL*) muscles in experimental microgravity conditions. Additionally, we carried out a comparative study using a well-established animal model (female Wistar and Sprague Dawley rats) of HU [29,30] at 1 g on the ground as an earth-based analog to spaceflight.

In both experimental animal models, we discovered that Homer crosslinking capacity is reduced specifically in postural muscle. To our knowledge, this is the first study proposing a shared molecular mechanism for skeletal muscle atrophy during unloading following real microgravity exposure and HU immobilization at 1 g on the ground.

## 2. Results

### 2.1. Microgravity Exposure Increases Homer1a and Decreases Homer2a/b Transcription in Antigravity Postural Mouse SOL Muscle

A total of 680 genes were found in previous experiments of gene array analysis of which two were Homer1 and Homer2, but not Homer3, whose expression was differentially regulated in muscle *SOL* of spaceflight mice housed on board the Bion-M1 (BF) biosatellite for 30 days in orbit [31].

To further investigate Homer-specific isoforms changes, the expression pattern of Homer short (Homer1a) and long (Homer2) isoforms were analyzed by quantitative PCR (qPCR) normalized using four different housekeeping genes. No significant differences were present in either muscle type between ground controls (BG vs. FC), excluding any side effects due to the animal housing within Bion-M1 biosatellite per se. More interestingly, qPCR data analysis showed that Homer1a and Homer2 were differentially regulated in *SOL* muscle from BF vs. BG. Notably, 30 days of microgravity exposure aboard Bion-M1 evoked a trend toward increased the transcription of Homer1a mRNA vs. BG (Figure 1A). In addition, a significant decrease in Homer2 expression was observed vs. the same ground control (Figure 1B).

No changes for Homer transcripts whatsoever were detectable in *EDL* muscle from FC, BG, and BF groups, suggesting at least for Homer genes “muscle-specific microgravity effects” in our experimental model.

Oligonucleotide primers used are listed in Table 1.

These results fully overlap with the results obtained by global gene array analysis showing that *homer1* gene transcription was 2.0 fold increased in BF mice *SOL* muscle compared to both ground, BG, and FC control groups (Appendix A), whereas *homer2* gene transcription was 3.6 fold decreased (Appendix A). In contrast, no changes were detected for either *homer1* or *homer2* gene transcription in the *EDL* muscle (Appendix A).

### 2.2. Microgravity Exposure Decreases Homer Protein at the NMJ Postsynaptic Microdomain in Mouse SOL Muscle

The subcellular localization and expression pattern of Homer proteins were determined at the NMJs by a high-resolution laser confocal microscope measuring the relative fluorescence intensity of immunostained postsynaptic membrane structures [16]. For this purpose, two different pan-Homer antibodies equally immunoreactive for all isoforms were used [32]. The two antibodies were previously raised in rabbit against bacterially expressed GST-Homer fusion proteins (Homer EVH1 domain) that included either the first 120 amino-terminal amino acid (RB-03) or the first 131 amino-terminal amino acids (RB-04) [32]. Briefly, anti-Homer antibodies staining in *SOL* muscle of BF (Figure 2A, upper right panel), as compared to BG (Figure 2A, upper left panel), highlighted a decreased fluorescence intensity (green signal) at the NMJ postsynaptic regions beneath the nicotinic acetylcholine receptor (nAChRs, red signal). Similar results were obtained using both RB-03 and RB-04 purified anti-Homer antibodies.

By contrast, no decrease in Homer immunofluorescence intensity was observed in the *EDL* muscle (antagonist shin muscle to calf SOL) using either anti-Homer antibodies (Figure 2A *EDL*, lower panel), although the slight differences observed in the expression pattern suggested a muscle adaptation to spaceflight.

The use of the longitudinal plane of NMJ would have been optimal for Homer protein subcellular localization analysis. However, due to the scarcity of muscle material from spaceflown mice, we were not able to prepare, stain, and study NMJ as necessary, for example, in laboratory animals [33]. Control immunostaining experiments with Homer pre-immune serum and/or with secondary antibodies alone did not reproduce the typical Homer immunostaining pattern described above, confirming the Homer antibody specificity (data not shown).

For data quantification, laser confocal Homer pixel intensity analysis was performed in the proximity of the NMJs postsynaptic region of BF vs. BG and FC (Figure 2B). No differences were detected in BG vs. FC control groups, excluding any side effects of the biosatellite housing at NMJ levels. Conversely, Homer signal levels were decreased in *SOL* muscle of BF compared to BG and FC control groups. Fluorescence intensity analysis also confirmed no changes in the *EDL* muscle from flown mice at the NMJ level.

The prominent changes in *SOL* and absence of change in *EDL* muscle support the notion of muscle-specific microgravity effects on Homer expressed at NMJ postsynaptic microdomains.

### 2.3. Microgravity Exposure Decreases nAChRs at the NMJ Level in Mouse SOL Muscle

Fluorescence intensity of α-Bungarotoxin (α-BTX) stained nAChRs was similar in BG and FC control groups, but it appeared to be decreased in *SOL* NMJs of BF mice, as compared to NMJs of BG and FC groups (Figure 3, upper panel). In contrast, no significant changes were observed in *EDL* NMJs from all groups (Figure 3, lower panel).

Thus, muscle-specific effects in nAChR distribution patterns were found in a postural (*SOL*) vs. non-postural muscle (*EDL*), suggesting that a muscle-specific adaptation/remodeling mechanism occurred at postsynaptic structures following 4 weeks of spaceflight.

### 2.4. One to 15 Days of Hindlimb Unloading Caused Transient Increase in Homer1a and a Long-Lasting Decrease in Homer2a/b mRNA Transcripts in Rat SOL Muscle

In order to dissect the direct effects of microgravity exposure from muscle unloading in Space, Homer short and long isoform expression patterns were further investigated in *SOL* muscle of female Wistar rats subjected to HU, an accepted ground-based disuse model in rodents. Short (1a) and long isoform (1b/c; 2a/b) mRNA transcripts for Homer were investigated in *SOL* from control (CON) and HU rats by real time qPCR. The time course of Homer1a (the dominant negative or inducible isoform), Homer1b,c and of Homer2a,b transcripts in *SOL* muscle from CON and HU rats was evaluated after 1, 4, 7, and 15 days of HU. Results demonstrated a progressive decrease in all Homer transcripts in the rat *SOL* muscle beginning on day 1 of HU. However, transcription of Homer1a showed a clear transient increase between 4–7 days of HU, whereas transcription of Homer2 decreased significantly over the entire 15 days of HU period (Figure 4A). Oligonucleotide primers used are listed in Table 2.

### 2.5. Muscle Weight/body Weight Ratio Data Changes during HU (rat)

Compared to CON female Wistar rats, normalized *SOL* muscle mass was reduced by about 25% after HU 4–7 days and 40% after the second week of HU (ANOVA *p* < 0.001) (Figure 4B). The CSA (cross-sectional area) analysis confirmed the absence of significant muscle atrophy after 1 day HU (Figure 4C) as already published for another animal group after 1 day HU [35]. CSA values of both slow or type I and fast or type II myofibers significantly decreased after 4, 7, and 15 days HU (Figure 4C), consistently with previously published data [36]. Three-week HU decreased CSA of *SOL* type-I myofibers to −52% [37].

This reduction largely mimics the decreased myofiber cross-sectional area of the BF *SOL* muscle, which was −53% for type-I, −44% for type-IIA, −50% for type-IIB, and −9% for type-IIX [31]. Considering the results altogether, it appears that Homer transcripts decrease before the occurrence of muscle atrophy. Once atrophy is established, Homer 1a levels are transiently upregulated, and Homer2 levels remain reduced.

Data are expressed as means ± SE. Statistical differences between groups were determined by an unpaired *t*-test. Within the HU group, One-Way Repeated Measures Analysis of Variance (ANOVA) analyzed comparisons between time points.

### 2.6. 3 Weeks of Hindlimb Unloading Decreased Homer Protein Expression and Subcellular Localization at the NMJ of rat SOL Muscle

The Homer protein isoform expression pattern was investigated in ***female Sprague Dawley*** rat *SOL* after 3 weeks of HU by Western blotting and immunohistochemistry.

In Figure 5A (upper panel, RB-03 immune serum individual animal muscle sample each line; middle panel, RB-03 affinity-purified anti-Homer antibodies), and Figure 5B (pooled samples, each animal group each line, *n* = 4), anti-Homer antibodies identified immunoreactive polypeptide bands with an apparent molecular weight of 45–48 kDa.

The predicted molecular weights (MWs) of all Homer long protein isoforms were clearly detected in the *SOL* muscle lysates of the CON group (Figure 5A) compared to the HU. As judged by α-tubulin, comparable total proteins were loaded for each animal muscle sample/group line (Figure 5A, lower panel). Brain total protein homogenate was used as Homer positive control.

We next checked whether changes in Homer protein expression pattern could be related to skeletal muscle inactivity/disuse. Thus, *SOL* homogenates from CON and HU rats were pooled and analyzed for pan-Homer (RB-03 affinity purified) antibodies (Figure 5B).

Densitometry analysis confirmed a significant decrease in Homer protein expression detectable in *SOL* muscle (Figure 5B right panel) of the HU rats. In particular, 45–48 kDa MW bands referable to Homer long isoforms were affected, confirming that Homer protein expression levels were more likely compromised by disuse following HU in the rat (Figure 5A and B left panel arrows).

A set of Homer immunofluorescence experiments were performed using a routine epifluorescence microscope to analyze Homer protein expression and subcellular localization in rat skeletal muscle. As expected, in CON *SOL* muscle, immunofluorescence analysis (Figure 5C) using affinity-purified RB-03 anti-Homer antibodies identified Homer protein antigens in two skeletal muscle subcellular compartments: at the Z-disc (the cross-striated pattern) and the NMJ (identified by α-BTX staining). The detailed analysis of anti-Homer immunoreactivity in *SOL* of HU vs. CON rats showed a decrease in Homer immunofluorescence staining at the NMJ.

### 2.7. Homer Protein Expression and Subcellular Localization in Homer2^−/−^ Mice

*SOL* muscle from WT and Homer2^−/−^ male C57BL/N6 mice were used to confirm Homer2 subcellular localization at the NMJ.

Figure 6A confirms the absence of Homer2 allele in the Homer2^−/−^ mouse (Mouse 1 and 3) vs. WT mouse (Mouse 2 and Mouse Control WT).

As shown in Figure 6B, in WT mice *SOL* muscle, anti-Homer antibodies recognized Homer antigens localized in two distinct and separated skeletal muscle subcellular compartments as already reported [16,32] in the proximity of the NMJ postsynaptic microdomain and at the Z-disc/costamere level resulting, the letter, in a cross-striated immunofluorescence pattern.

In *homer2*^−/−^ mice *SOL* muscle, Homer fluorescence immunoreactivity was highly decreased at the NMJ postsynaptic microdomain, whereas the cross-striated pattern persisted in this postural muscle. Since our anti-Homer antibodies were isoform unselective [32], the remaining faint immunofluorescence pattern at the NMJ postsynaptic microdomain and the immunofluorescence pattern at the Z-disc/costamere in *homer2*^−/−^ mice *SOL* most likely represented Homer1 proteins [32]. Thus, our findings confirmed anti-Homer antibody specificity in immunohistochemistry experiments and suggested that Homer2 represented the predominant NMJ Homer protein isoform, whereas the bulk of Homer1 is particularly confined at the muscle Z-disc/costamere subcellular structure.

## 3. Discussion

The present data support a model in which postsynaptic downregulation of crosslinking Homer gene was an early event of slow-twitch muscle atrophy following unloading. While there are aspects of this response that are unique to microgravity, substantially similar events also occurred with muscle unloading on Earth, suggesting reciprocal expression of Homer isoforms (Homer1a vs. Homer 2) as common and key signaling mechanisms toward disuse atrophy.

Crosslinking forms of Homer (Homer 1b/c and Homer2a/b) were downregulated, and the form of Homer that disrupts Homer crosslinking (Homer1a) was either transiently or persistently upregulated in disuse. The consequent reduction in postsynaptic Homer protein was evident by immunohistochemistry and was closely associated with reduced expression of nAChRs. The reduction in postsynaptic crosslinking Homer is likely to have a functional impact on NMJ function. Homer proteins bind and couple the actions of multiple proteins that are essential for NMJ function, including the RyR and IP3R [19,22,38,39] Homer proteins also bind NFAT [16,40], which is important for signaling to local nuclei and maintenance of muscle gene expression. Changes in Homer expression appeared to be specific to functional muscle subtypes; in the present and published work, reductions in Homer transcription and protein expression were specific to the postural slow-twitch muscle that typically mediate antigravity support and must be constitutively active, while changes in Homer were not seen in fast-twitch muscles that can normally be inactive for extended periods of time. A decrease in Homer immunofluorescence intensity was present in *SOL* muscle at the NMJ postsynaptic microdomain following 30 days of microgravity exposure on board the biosatellite Bion-M1 and after 21 days of HU on ground. Whereas at the transcriptional level, although only transiently in the HU, an increase in Homer1a and a decrease in Homer2a/b mRNA transcripts were present in *SOL* muscle from both investigated experimental models. These changes were not detected in the *EDL* muscle observed as far as Homer protein subcellular localization and mRNA transcripts are concerned.

Despite similarities to hindlimb unloading, real microgravity apparently showed specific effects on Homer gene expression. Both the absence of gravity and/or simulated microgravity models negatively regulated Homer protein expression and NMJ subcellular localization in a muscle-specific and time-dependent fashion. The *SOL* muscle appeared to be one key postural limb muscle, where the transcripts for the Homer2 longer isoforms appeared early and permanently downregulated in both spaceflight and similar conditions, as studied in our experimental models. Conversely, the Homer1a dominant negative form, whose final effects were additive to Homer2 downregulation, appeared to be persistently upregulated only in spaceflown mice *SOL*. Although different experimental conditions and methods sensitivity might have influenced data quality outcome to some extent, our findings altogether clearly point to a microgravity-related increased Homer1a expression.

Present and previous observations [16,17] suggest that Homers are key players in skeletal muscle adaptation and plasticity. In particular, Homer1a and Homer2 are required for trophic homeostasis of slow-twitch muscle in various gravitational conditions, at least in small animals (mice, rats). Further evidence supporting this hypothesis emerged from the comparative analysis of the mainly slow-twitch *SOL* versus the mainly fast-twitch *EDL* muscle from rat hind limbs. Moreover, the current understanding of skeletal muscle atrophy has been confirmed by the fact that slow-twitch muscles, such as the *SOL*, are more sensitive to unloading conditions, i.e., during long-term bed rest, HU or exposure to microgravity, than fast-twitch skeletal muscles, such as the foot sole plantaris and calf medial gastrocnemius [41]. However, molecular mechanisms responsible for the differential muscle-type unloading response are not yet fully understood, and further experiments are needed to demonstrate the functional role of Homer-specific isoforms.

We recently reported that Homer 2 expression decreased after 60 days of bedrest (BR) muscle disuse and that the application of resistive vibration exercise (RVE,) as a countermeasure intervention was able to counteract or even increase Homer2a/b transcription and protein expression at the NMJ postsynaptic subcellular microdomain suggesting peculiar mechanosensitive mechanisms for Homers in otherwise disused human muscle [16]. Thus, Homer2a/b expression was characteristic of the BR-biopsy and BR-group, whereas Homer1b/c level was identical irrespective of the BR-biopsy and BR-groups.

The transition of Homer-specific isoforms in rat slow-twitch skeletal muscle has been proposed in two different and well-established experimental animal models of muscle wasting/atrophy, i.e., denervation and HU skeletal muscles [17]. Similar to human skeletal muscle, no change in Homer1b/c was present in skeletal muscle of adult rats up to 14 days of in situ denervation. By contrast, Homer2a/b was significantly decreased by as much as 70% and 90% at 7 and 14 days of denervation, respectively, which paralleled the reduction in muscle mass. Similarly, 7 days of HU decreased Homer 2 by 70% [16,17].

Despite a large body of evidence indicating the deleterious effects of chronic unloading to muscle NMJ structure and function, conclusive studies concerning the molecular mechanisms of NMJ adaptation and plasticity following long-term muscle unloading on the ground and/or chronic exposure to microgravity are scarce in the literature. Deschenes et al. [15] reported an increased NMJ structural/morphological remodeling in mouse skeletal muscle after 16 days ISS microgravity. Moreover, there are several lines of evidence suggesting that skeletal muscles, and in particular lower limb antigravity postural muscles, undergo several adaptive processes during spaceflight missions mostly characterized by structural and metabolic changes [41,42,43], which are largely responsible for altered skeletal muscle mass, structure, and function.

By using mostly mice and rats, as experimental animal models, it was shown that microgravity exposure, even for a short period of 2–3 weeks, alters skeletal muscle structure and function. Some types of muscle, however, undergo a greater degree of atrophy than others. In some studies, for instance, the high degree of muscle atrophy well correlates with the presence of the slow type myosin heavy chain and with a greater proportion of slow-twitch fibers. In other reports, however, muscle atrophy was similarly described in both slow- and fast-twitch myofibers [42]. An attractive explanation for the relative atrophic response elicited in different muscle types is that microgravity might induce a shift in “neuronal recruitment patterns” of motor units [42,44]. Thus, nerve ending remodeling may play an additional and synergistic role during spaceflight, and altered neuron-derived specific signals (electrical and/or chemical transmitters) [45,46] may contribute to the microgravity effects at the level of single skeletal muscle fibers.

The most relevant changes at the presynaptic level during spaceflight microgravity conditions are a decreased number of synaptic vesicles and neurotransmitter content, degeneration of axon terminals, and axonal sprouting [10,11,12,13,47], which in part also occur in aging muscle [48,49]. Interestingly, in all models of disuse-induced muscle atrophy (bed rest, denervation, and limb casting/immobilization), the NMJ undergoes disruptive structural adaptations similar to those observed during exposure to microgravity [11,42]. Nevertheless, even though there are several studies in the literature which have investigated NMJ adaptations during spaceflight, it is worth noting that most of them were conducted at the cellular, structural, and ultrastructural level, thus lacking molecular evidence from spaceflown small animals or even astronauts.

Given the unique ability of Homer long isoforms to provide a complex scaffold network for signaling proteins, the functional role of Homer proteins at the NMJ postsynaptic microdomain is quite intriguing. In a previous study, we provided compelling evidence suggesting that Homer2 isoforms may regulate the calcineurin/NFATc signaling pathway in human [16] and rat [17] skeletal muscle.

In the present study, 30 days of microgravity exposure resulted in the upregulation of the short- and downregulation of the long-Homer isoform in slow-twitch muscles, whereas muscle loading even for a short period from landing to muscle dissection (approximately 24–28 h post-landing) was not sufficient to rescue microgravity adaptations with respect to Homer transcription.

These results are not only consistent with those reported in the literature showing that Homer2 participates in the control of ubiquitination and consequent proteolysis via transcriptional downregulation of MuRF1, Atrogin 1, and Myogenin in slow-twitch muscles [17] but also suggest a crucial role of the Homer protein family in the onset of unloading-induced muscle atrophy. In fact, Homer2 transcript downregulation occurs synchronously to Atrogin-1 and FoxO3 upregulation, which was detected in the soleus muscle only after 24 h unloading [50].

We conclude that Homer-specific isoforms are highly regulated by muscle activity/inactivity at the NMJ and that Homer isoform sensitivity to real microgravity exposure is muscle-type specific. Altered NMJ molecular structure, shown by differential subsynaptic Homer isoform expression, suggests a key signature to muscle atrophy following chronic muscle unloading in a spaceflight analog with rats on Earth gravity (HU) and/or real microgravity exposure in 30 days spaceflown mice.

However, the present work, although indicative of a molecular mechanism underlying disuse-induced muscle atrophy, presents two unavoidable limitations. The first is represented by the reduced number of available flown animals imposed by the small animal modules available during spaceflights missions. The second limitation pertains to the use of two different strains of rats instead of mice for the comparative disuse experiments (or spaceflight analog) on ground (hindlimb unloading) due to ethical restrictions. Thus, further studies are needed to confirm the present data and hypothesis and to further investigate downstream molecular signaling pathways linked to postsynaptic reciprocal Homer isoforms expression in various conditions of skeletal muscle disuse.

## 4. Materials and Methods

### 4.1. Space Flown Animals

C57BL/N6 male mice (22–25 g body weight) were purchased from the Animal Breeding Facility Branch of Shemyakin and Ovchinnikov Institute of Bioorganic Chemistry, Russia. Mice were then transported to the animal facility of Moscow State University, Institute of Bioengineering, for preadaptation training and selection in the laboratory setting on the ground before flight. In all experiments, mice between 19 and 20 weeks old were used. Animals underwent a preflight animal training and a preselection program (e.g., preadaptation to standard laboratory cage conditions and familiarization of individual mice groups compliant for housing in smaller animal flight habitats used for spaceflight missions [31,51]. Mice were then randomly divided into 3 groups: 1. Mice to be flown aboard the Bion-M1 capsule exposed for 30 days to microgravity (Bion-M1 Flown = BF); 2. Mice housed for 30 days in the same biosatellite on ground under the same bio-parameters, i.e., the number of animals per group and identical housing conditions in a Bion-M1 used habitat (Bion-M1 Ground = BG); and 3. Mice housed in the animal facility in standard cage conditions during the space flight (FC = Flown Control). A general study protocol of the BION-M1 space mission, including murine animal training and selection, has been published elsewhere [28]. The study was approved by IACUC of MSU Institute of Mitoengineering (Protocol No–35, 1 November 2012) and of Biomedical Ethics Commission of IBMP (protocol No–319, 4 April 2013) and conducted in compliance with the European Convention for the Protection of Vertebrate Animals used for Experimental and Other Scientific Purposes [28].

### 4.2. Sample Preparation and Transportation

After landing, mice were transferred in the Bion-M1 housing device within 12–14 h from the landing site (Kazakhstan) to Moscow. Our Russian contracted partners on-site (Institute of Biomedical Problems, IMBP, Moscow, Russia, contract #Bion-M1/2013 between RF SRC-IMBP and the Charité Berlin, Germany) accomplished all operational support with preflight animal handling, post-flight animal tissue dissection, and sample freezing. All frozen samples were delivered from IMBP Moscow on dry ice by World Courier GmbH, Berlin, Germany, via temperature-controlled express delivery parcels to the Charité Berlin, Germany. The samples were stored at −80 °C until they were used for the experiments.

### 4.3. Hind Limb Unloading (HU) Experimental Animals and Animal Care

Due to ethical and technical limitations to using mice in experiments of HU, adult rats were used for HU vs. normal cage control. For experiments reported in Figure 4, a first group of 6-week-old female Wistar rats (140–160 g of body weight) housed in a cage individually were used for HU (*n* = 20) and CON (*n*= 15) experiments. The HU protocol [17] was performed according to the recommendations provided by the European Convention for the Protection of Vertebrate Animals used for Experimental and Scientific Purposes (Council of Europe No. 123, Strasbourg, 1985) and authorized by the Animal Ethics Committee (OPBA) of the University of Padua and the Italian Health Ministry (103/2007B). HU muscles were unloaded by using the tail-suspension experimental model (29, 30). Each animal was weighed before and after the HU period. HU rats were sacrificed after 1 (*n* = 6), 4 (*n* = 5), 7 (*n* = 5), and 15 (*n* = 4) days of HU. Soleus muscles were excised, weighed, and snap-frozen in liquid nitrogen and then stored at −80 °C until used for the experiment.

For experiments reported in Figure 5, a second group of 8-week old female Sprague Dawley HU (*n* = 6) and CON (*n* = 6) rats were used, purchased from the Experimental Animal Center of Beijing University (body weight ranged from 215 to 235 g). After one week of adaptation to standard laboratory cages (*n* = 2, each cage), twelve animals (*n* = 12) were randomly selected and divided into two groups (*n* = 6, each): Hind limb unloading (HU), and a control (CON) group. In HU group, rats were subjected to the tail suspension method [29,30] for the duration of 21 days. Both HU and CON groups were subjected to the same nursery/housing conditions with 12 h dark-light cycles and food and water ad libitum for 21 days in the animal facility of the Department at Beijing University, China. Animal treatment and care were according to Regulations for the Administration of Affairs Concerning Experimental Animals promulgated by Decree No.2 of the State Science and Technology Commission of China and the Guiding Principles for the Care and Use of Animals approved by the Beijing Government. All protocols were approved by the Animal Care Committee of Beijing University, China [37].

### 4.4. Homer2^−/−^ Mice

Male C57BL/N6 WT and *homer2^−/−^* mice were kindly provided by the laboratory of one co-author (P.W.), Department of Neuroscience, Johns Hopkins University School of Medicine [52].

### 4.5. Homer2 Mice Genotyping

Primers sequences: 147: 5′ GTG GGT GGC CTA GAA ATC AT 3′; 148: 5′ CTT CCG GAA CCC TTC ATC TT 3′; 149: 5′ CTC GCT ACC TTA GGA CCG TTA 3′. Product size: Primers 147 and 148 make a wild-type band at 321bp. Primers 147 and 149 make a fainter mutant band around 432bp.

### 4.6. Immunohistochemistry

Single muscle cryosections (8 μm thickness) from *SOL* and *EDL* were cut (Leica Microsystems, Bensheim, Germany), mounted on a microscope slide (Superfrost plus, VWR International, Radnor, PA, USA), and stored frozen in sealed plastic boxes (−80 °C) until further use. From the set of cross-cut frozen tissue cryosections available, we selected those showing NMJ structures as identified by NMJ-specific marker alpha-Bungarotoxin (αBTx) prior to Homer immunohistochemistry. Thus, tissue cryosections were double-labeled with affinity-purified anti-pan Homer antibodies (16, 32) and Alexa-555-conjugated αBTX, a specific marker of NMJ acetylcholine receptors. Primary anti-Homer antibodies were then visualized by using Alexa-488 conjugated goat-anti-rabbit secondary antibodies (Invitrogen, Carlsbad, CA, USA).

Immunostained cryosections were analyzed using an epifluorescence microscope (Axioplan; Zeiss, Oberkochen, Germany) equipped with a Cool Snap digital camera (Visitron Systems GmbH, Puchheim, Germany). Digitized images were processed by using MetaVue software (release # 2.61.1537) (Meta Series 7.5.6; system ID: 33693; Molecular Devices, Sunnyvale, CA, USA). Tissue sections were also inspected using a three-channel confocal laser scanning microscope (TCS SP-2, LEICA Microsystems, Bensheim, Germany) at standardized image settings [16], and all digitized images were analyzed using the LEICA confocal custom software.

The immunopatterns of Homer proteins in cryosections were determined by measuring the relative fluorescence intensity at postsynaptic membrane structures as previously described [16]. Briefly, the area pixel intensity of a selected region of interest (ROI) within the subsarcolemmal area in a muscle fiber of about 1.000 μm^2^ was measured in digitized confocal image scans and expressed as arbitrary units (range 0–255 a.u.). At least twenty NMJs were measured in each animal and group using tissue cryosections from at least three independent immunostaining experiments.

Changes in Homer immunofluorescence intensity were determined by area-based pixel intensity measurements between individual animals from each group (BF, BG, FC, *n* = 2 each).

Immunostaining experiments with secondary detection antibodies only (omitting 1st antibody) were used as internal negative controls (data not shown). The number of NMJ/animals analyzed for a specific experimental purpose was ≥12 for mice and ≥20 for rats.

### 4.7. Preparation of Muscle Homogenates

Snap frozen *SOL* and *EDL* muscle samples from mice and rats were placed in RIPA buffer (50 mM Tris-HCl, pH 7.4, 150 mM NaCl, 1% NP-40, 0.5% sodium deoxycholate, 0.1% SDS, 2 mM MgCl_2_ in the presence of the protease inhibitor cocktail) and each sample was homogenized using a glass Dounce homogenizer with 20 strokes (16). Muscle homogenates were then centrifuged at 14,000× *g* for 15 min at 4 °C in order to obtain the soluble fraction by removing the insoluble debris [16]. Aliquots were stored at−80 °C until used. Protein concentration was determined by using a colorimetric assay kit (Pierce, ThermoFisher Scientific, LSR Rockford, IL, USA).

### 4.8. Sodium Dodecyl Sulphate (SDS)-Polyacrylamide Gel Electrophoresis (PAGE) and Western Blot Analysis

Each muscle sample was diluted to a final concentration of 4 mg/mL in SDS-containing loading buffer. Proteins were resolved by electrophoresis in SDS-containing running buffer (Bio-Rad Laboratories, Hercules, CA, USA), transferred to nitrocellulose membranes (blots), and incubated overnight with TBS-T (Tris-buffered saline, pH 7.5, 0.2% Tween^®^20) supplemented with 4% non-fat dry milk at 4 °C. Blots were probed with the primary antibody (1:500 dilutions) in TBS-T solution supplemented with 4% non-fat dry milk. After extensive washing, blots were incubated for 1 h with either anti-rabbit or anti-mouse alkaline phosphatase-conjugated secondary antibodies (1:500 dilution) in TBS–T solution. Secondary antibodies were detected by using the alkaline phosphatase substrate BCIP/NBT [16]. Protein content was quantified by densitometry using a GS-800 calibrated imaging densitometer (Bio-Rad, Quantity-One™ software Quantity-One-4.5.2, Biorad). Alpha-tubulin was used as a protein loading marker [16].

### 4.9. RNA Extraction and Sample Target Preparation

Total RNA was isolated from mouse *SOL* (*n* = 3) and *EDL* (*n* = 3) of each space experimental group (BF, BG, and FC) and from rat *SOL* and *EDL* of each ground-based experimental group (CON *n* = 15, HU day1 *n* = 6, day 4 *n* = 5, day 7 *n* = 5, day 15 *n* = 4) using an RNeasy Micro Kit (Qiagen, Hilden, Germany). Frozen tissue samples were ground to a fine powder under liquid nitrogen. A homogeneous lysate was achieved by adding lysing buffer and pipette the lysate up and down 10 times through a sterile syringe needle. Tissue lysate was centrifuged, and the supernatant was used for RNA phenol/chloroform extraction. A single-step method of RNA isolation by acid guanidinium thiocyanate–phenol-chloroform extraction was applied. The aqueous layer was mixed with an equal volume of 70% ethanol, and total RNA was extracted using RNeasy spin columns, according to the manufacturer’s protocol. 2100 Bioanalyzer (AGILENT Technologies, Santa Clara, CA, USA) was used to check RNA integrity. The amplification and labeling of the RNA samples were carried out according to the manufacturer’s instructions (Affymetrix, Santa Clara, CA, USA). Briefly, total RNA was quantified and checked by analysis on a LabChip (BioAnalyzer, AGILENT Technologies). The GeneChip^®^ 3′ IVT Express Protocol is based on the Eberwine or reverse transcription method (in vitro transcription, IVT). Starting from 100 ng total RNA, first-strand DNA was synthesized, containing a T7 promoter sequence and then converted into double-stranded DNA. The double-strand DNA served as template in the subsequent in vitro transcription (IVT) reaction. This amplification step generated Biotin-labeled complementary RNA (cRNA). After cleanup, the biotin-modified RNA was fragmented by alkaline treatment. Fifteen micrograms of each cRNA sample was hybridized for 16 h at 45 °C to an Affymetrix GeneChip Mouse 430A 2.0 Array [31]. Arrays were washed and stained with streptavidin–phycoerythrin solutions using a fluidics station according to the protocols recommended by the manufacturer. Finally, probe arrays were scanned at 1.56-μm resolution using the Affymetrix GeneChip System confocal scanner 3000. The Affymetrix Mouse Genome 430A 2.0 Array includes 22,600 probes sets to evaluate the expression level of more than 14,000 well-characterized mouse genes [31].

### 4.10. Quantitative Polymerase Chain Reaction (qPCR) Analysis

Quantitative PCR was performed by the SYBR Green method as previously described (16). Briefly, 400 ng of RNA were converted to cDNA by using random hexamers and SuperScript^®^ VILO™ (ThermoFisher Scientific Inc., Waltham, MA, USA), following the manufacturer’s instructions. Specific primers that were used for qPCR were previously published elsewhere [16,17,34] or newly designed using Primer3 software (http://frodo.wi.mit.edu/, Whitehead Institute for Biomedical Research) (accessed on 8 July 2010). Their thermodynamic specificity was determined using BLAST sequence alignment (NCBI) and vector NTI^®^ software (Vector NTI Advance^TM^ 11.0, December 2008, Invitrogen Corporation). The oligonucleotide primers used are listed in Table 1 and Table 2.

The reaction mix consisted of 10 μL of 2x iQ SYBR Green Supermix (Bio-Rad), 0.3 pmol/μL primers, 8 ng of cDNA, and DNase/RNase-free water up to 20 μL. The PCR parameters were initial denaturation at 95 °C for 30 s followed by 40 cycles of 10 s at 95 °C and 30 s at the corresponding annealing temperature (53–59 °C) for the acquisition of a fluorescence signal. A melting curve was generated by the iQ5 software (Bio-Rad iQ5- Standard Edition, Version 2.0.148.60623) following the end of the final cycle for each sample by continuously monitoring the SYBR Green fluorescence throughout the temperature ramp from 65 °C to 99 °C in 0.5 s increments. All samples were run in triplicate, in parallel for each individual muscle sample, and simultaneously with RNA-negative controls. Cyclophilin A (PPIA), pyruvate carboxylase (PCX), glyceraldehyde 3-phosphate dehydrogenase (GAPDH), and Beta-actin (ACTB) were tested as candidate reference genes for mouse samples (Table 1), whereas Cyclophilin A (PPIA), Beta-actin (ACTB), and TATA-Box Binding Protein1 (TBP1) were tested for rat samples (Table 2). PPIA was used to normalize Ct values by ΔCt method in all experiments. Same data trends were obtained if other reference genes were used (data not shown).

## Figures and Tables

**Figure 1 ijms-23-00075-f001:**
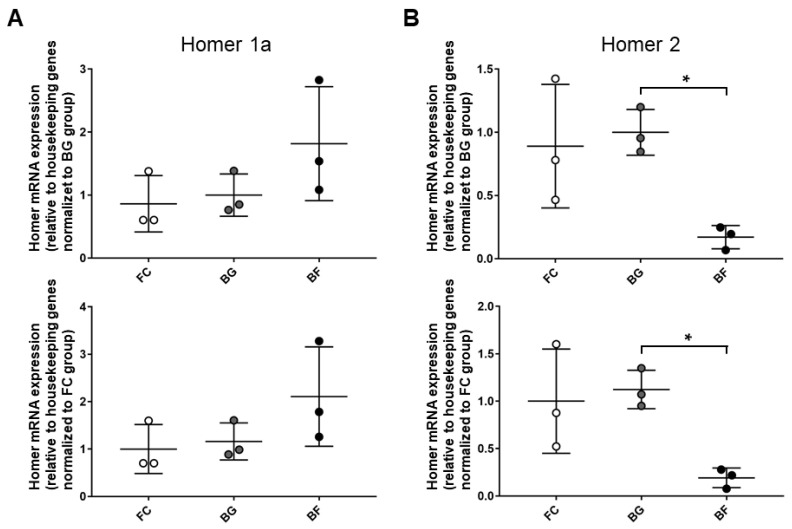
Homer1a and Homer2 genes are differentially regulated in *SOL* muscle of Bion-M1 flown mice. Results qPCR data analysis. (**A**) An increased, albeit not statistically significant, value of Homer1a mRNA expression was observed in *SOL* muscle of BF vs. BG (61.5%; *p* ≤ 0.789) and BF vs. FC (110%; *p* ≤ 0.177). (**B**) A significant decrease in Homer2 mRNA expression was observed in *SOL* muscle of BF vs. BG (−84%; *p* ≤ 0.021) and BF vs. FC (−81%; *p* ≤ 0.0664). mRNA levels were normalized to housekeeping genes by the delta Ct method and then normalized to CON (BG upper panel and FC lower panel), which was arbitrarily set to 1. Data are expressed as means ± SE. For each experimental group, *n* = 3 male C57BL/N6 mice (19–20 weeks old). Statistical differences between groups were determined by an unpaired *t*-test. * = indicates significant difference vs. CON (ANOVA *p* < 0.001).

**Figure 2 ijms-23-00075-f002:**
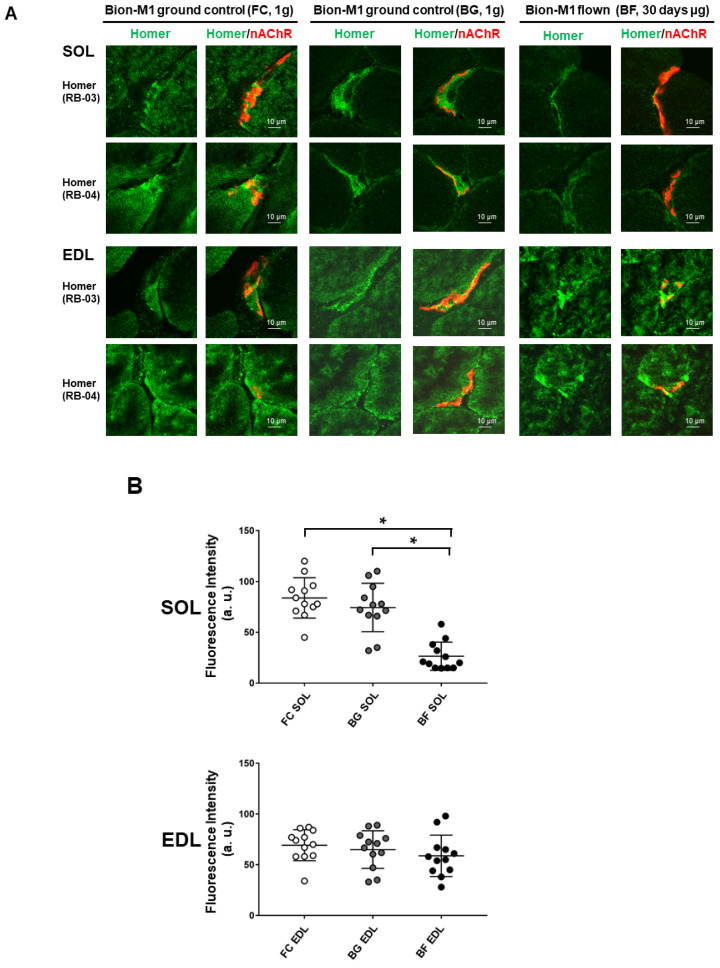
Confocal analysis of Homer immunostaining at the NMJ of *SOL* and *EDL* muscles in Bion-M1 flown mice (BF) vs. ground controls (FC and BG). (**A**) A reduction in Homer immunofluorescence signal adjacent to α-BTX stained nAChRs (NMJ postsynaptic microdomain) was observed in Bion-M1 flown mice compared to the Bion-M1 ground controls in *SOL* but not in *EDL* muscle. (**B**) Quantification of Homer immunofluorescence intensity at the NMJ of *SOL* (upper panel) and *EDL* (lower panel) muscles in Bion-M1 flown mice vs. ground controls. A significant decrease in Homer fluorescence intensity was detected in *SOL* of BF (*p* ≤ 0.001) but not in *EDL* muscle compared to ground controls BG and FC. Data are expressed as means ± SE. For each experimental group *n* = 2 male C57BL/N6 mice (19–20 weeks old). Statistical differences between groups were determined by an unpaired *t*-test. * = indicates significant difference vs. CON (ANOVA *p* < 0.001).

**Figure 3 ijms-23-00075-f003:**
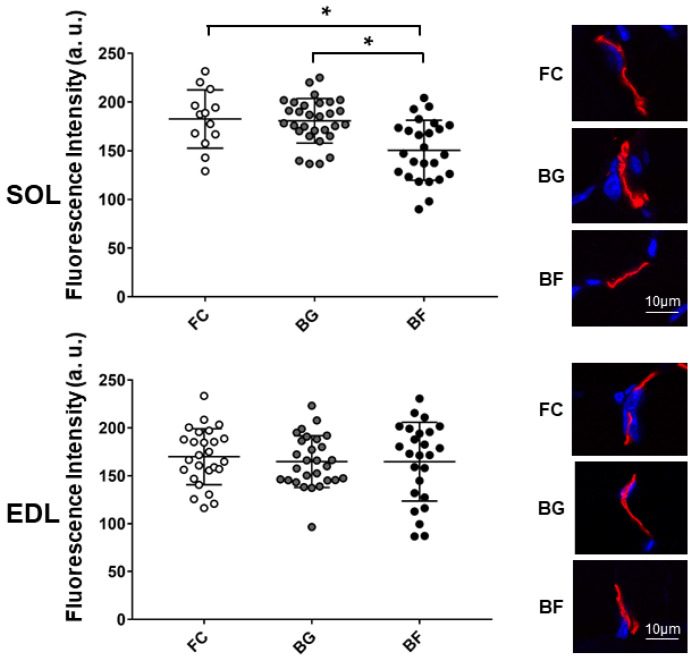
Confocal fluorescence intensity analysis of junctional nAChRs in *SOL* and *EDL* muscles of Bion-M1 flown (BF) vs. two Bion-M1 ground controls mice (FC, BG). A significant decrease in pixel fluorescence intensity was measured in *SOL* muscle (*upper panel*) of Bion-M1 flown mice (BF vs. BG, *p* ≤ 0.001; BF vs. FC, *p* ≤ 0.004) but not of *EDL* muscle (*lower panel*) (BF vs. BG, *p* ≤ 0.8929; BF vs. FC, *p* ≤ 0.6051) compared to the BG and FC muscle controls. Upper (*SOL*) and lower (*EDL*) graph, representative images of α-BTX stained nAChRs in BG, BF, and FC groups. No differences were observed between BG and FC control muscles. Data are expressed as means ± SE. For each experimental group *n* = 2 male C57BL/N6 mice (19–20 weeks old). Statistical differences between groups were determined by unpaired *t*-test. * = indicates significant difference vs. CON (ANOVA *p* < 0.001).

**Figure 4 ijms-23-00075-f004:**
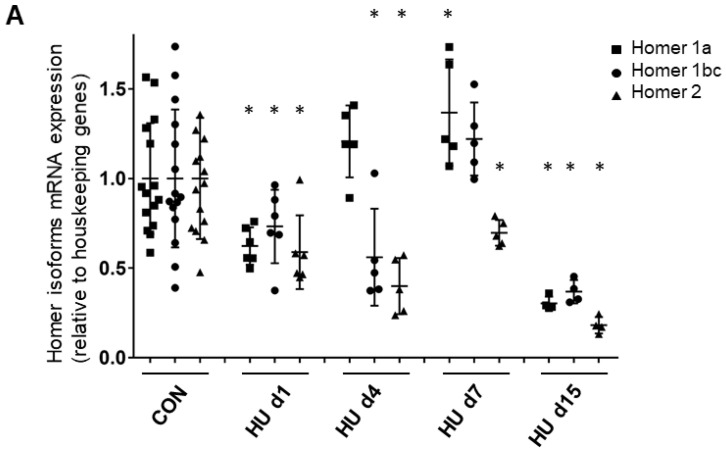
Time course of Homer 1a, 1b/c, and 2 isoform expression and morphometry analysis in rat *SOL* muscle. (**A**) Real-time qPCR analysis of ambulant CON vs. HU. A decrease in all Homer isoform transcripts was observed after 1 day of HU, which continued to decrease after 2 weeks: Homer1a −70% (*p* ≤ 0.0038), Homer1b,c −63,2% (*p* ≤ 0.0052), Homer2 −82% (*p* ≤ 0.0002). Only between day 4 and day 7, there was a transient increase in Homer1a isoform + 37% (*p* ≤ 0.032), but not of Homer1b,c + 22.1% (*p* ≤ 0.2417) compared to the CON group. mRNA levels were normalized to peptidylprolyl isomerase a (PPIA) gene (also known as cyclophilin A) by the delta Ct method and then normalized to CON samples, which were arbitrarily set to 1. The number of animals analyzed (6-week-old Wistar rats) was 15 for the ambulant group (CON) and 20 for the HU group. The HU animals analyzed for each experimental time point were 6 (day 1 HU, HU d1), 5 (day 4 HU, HU d4), 5 (day 7 HU, HU d7), 4 (day 15 HU, HU d15). (**B**) *SOL* muscle weight (MW)/body weight (BW) ratio. Compared to CON animals, there was a significant decrease of 25% after 4–7 days and 40% at day 15 of HU in female Wistar rats. These data showed that muscle mass decreased significantly from day 4 (ANOVA *p* < 0.001). A further significant reduction in SOL mass was observed between HU d7 and HU d15 groups (*p* = 0.04). The number of animals analyzed (6-week-old Wistar rats) was 13 for the ambulant group (CON) and 19 for the HU group. The HU animals analyzed for each experimental time point were 6 (day 1 HU, HU d1), 4 (day 4 HU, HU d4), 5 (day 7 HU, HU d7), 4 (day 15 HU, HU d15). (**C**) *SOL* slow (upper panel) and fast (lower panel) type myofiber CSA. Compared to CON animals, there was a significant decrease of −26.6% after 4 days, −45.55% after 7 days, and −53.53% after day 15 in slow, and a significant decrease of −26.5% after 4 days, −35.52% after 7 days, and −41.76% at day 15 in fast type myofibers of HU female Wistar rats. The number of animals analyzed (6-week-old Wistar rats) was 9 for the ambulant group (CON) and 19 for the HU group. The HU animals analyzed for each experimental time point were 6 (day 1 HU, HU d1), 4 (day 4 HU, HU d4), 5 (day 7 HU, HU d7), 4 (day 15 HU, HU d15). Each dot corresponds to the average CSA values calculated on 100 myofibers from two photographic fields for each muscle. * = indicates significant difference vs. CON (ANOVA *p* ≤ 0.001); ** = indicates significant difference vs. CON and HU day 1 and days 4 (ANOVA *p* ≤ 0.001).

**Figure 5 ijms-23-00075-f005:**
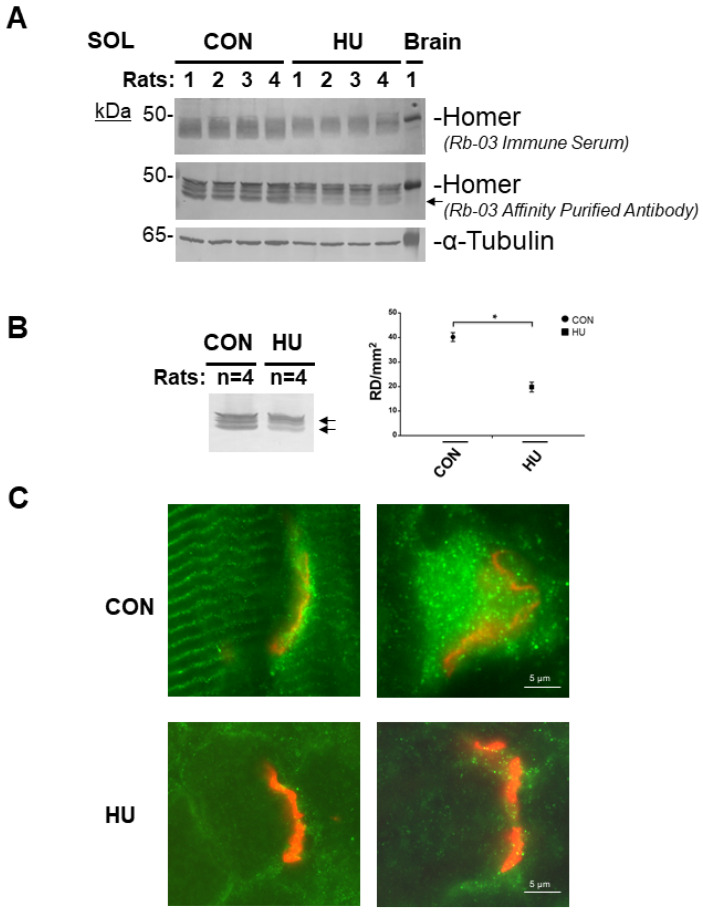
Homer protein expression analysis in rat *SOL* muscle after 3 weeks of HU. (**A**)**.** Western blot analysis of individual *SOL* muscle from ambulant (CON, *n* = 4) and hind limb suspended 8-week-old female Sprague Dawley rats (HU, *n* = 4). In reducing experimental conditions, anti-Homer antibodies identified several immunoreactive bands with an apparent molecular weight of 45 to 48 kDa; Homer predicted molecular weight. Brain was used as Homer protein positive control. α-tubulin (lower panel) was used as protein loading control. (**B**)**.** Left panel, Western blot analysis of pooled female Sprague Dawley rat *SOL* muscles from each experimental group. Right panel, Homer Densitometry (Reflexive Density, RD) analysis. Compared to CON, in HU group, after three weeks of unloading, there was approx. a 50% decrease in Homer total proteins (arrows). (**C**)**.** Homer immuno-epifluorescence analysis of female Sprague Dawley rat *SOL* NMJ of CON vs. HU. Representative merged images double-staining experiments with anti-Homer antibodies (green) and α-BTX (red) on longitudinal sections. Homer fluorescence immunoreactivity observed surrounding α-BTX-positive areas (NMJ postsynaptic microdomain) was less evident in 3-weeks HU compared to CON. In B, data are expressed as means ± SE. Statistical differences between groups were determined by unpaired *t*-test. * = indicates significant difference vs. CON (ANOVA *p* < 0.001).

**Figure 6 ijms-23-00075-f006:**
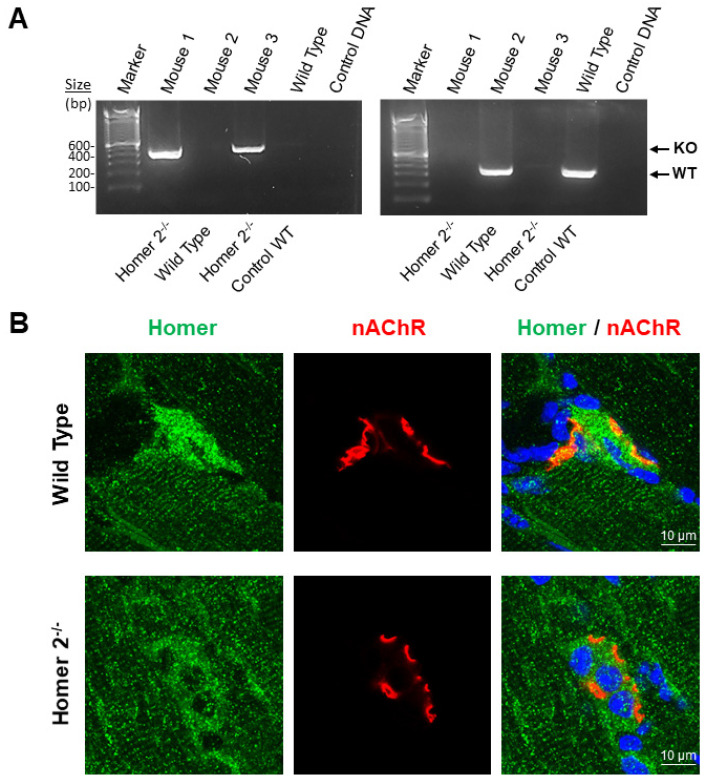
*homer2*^−/−^ genotyping and Homer confocal analysis in *SOL* muscle of WT and Homer2^−/−^ mice. (**A**). Agarose gel amplified cDNA fragment in WT and *homer2*^−/−^ male C57BL/N6 mice (19–20-week-old). As expected, a band of 321bp was detected in wild-type mice using a combination of primers 147 and 148 (WT), and a band around 432bp was detected in *homer2*^−/−^ mice using primers 147 and 149 (KO). (**B**)**.** Representative merged images of double staining experiments carried out with anti-Homer antibodies (green) and α-BTX (red) on *SOL* muscle longitudinal sections of WT and *homer2*^−/−^ male C57BL/N6 mice. Homer fluorescence immunoreactivity observed surrounding α-BTX-positive areas (NMJ postsynaptic microdomain) of WT mice was not detected in *homer2*^−/−^mice. No differences were observed at the Z-disc/costamere level (cross-striated immunofluorescence pattern) between WT and *homer2*^−/−^ mice.

**Table 1 ijms-23-00075-t001:** Primer sequences for mouse tissue qPCR analysis.

Gene	Primer	Sequence
Homer1a	forward reverse	5′-GAAGTCGCAGGAGAAGATGG-3′ 5′-GAACTTCCATATTTATCCA-3’
Homer2a/b	forward reverse	5′-CAGTGTATGTGACTCTCCAGCAG-3′ 5′-CTTTGTGGTTGACAATGTCATG-3′
PPIA	forward reverse	5′-AGCATGTGGTCTTTGGGAAGGTG-3′ 5′-CTTCTTGCTGGTCTTGCCATTCC-3′
PCX	forward reverse	5′-CCTCTCAGAGCGAGCAGACT-3′ 5′-TAGGGAAGCCGTAGGTGTTG-3′
GAPDH	forward reverse	5′-CACCATCTTCCAGGAGCGAG-3′ 5′-CCTTCTCCATGGTGGTGAAGAC-3′
ACTB	forward reverse	5′-CAAACATCCCCCAAAGTTCTAC-3′ 5′-TGAGGGACTTCCTCTAACCACT-3′

**Table 2 ijms-23-00075-t002:** Primer Sequences for Rat Tissue qPCR Analysis.

Gene	Primer	Sequence
Homer1a	forward reverse	5′-CCAGAAAGTATCAATGGGACAGATG-3′ 5′-TGCTGAATTGAATGTGTACCTATGTG-3′ [34]
Homer1b/c	forward reverse	5′-GTGAAGCAGTGGAAGCAACA-3′ 5′-CAGCTCCTGCACTGTCTGAC-3′ [34]
Homer2a/b	forward reverse	5′-TCTTGCTTCTCTGGCTTTGT-3′ 5′-CTGCGTAAACGGCTAAGGTA-3′ [17]
PPIA	forward reverse	5′-AGCATGTGGTCTTTGGGAAGGTG-3′ 5′-CTTCTTGCTGGTCTTGCCATTCC-3′
ACTB	forward reverse	5′-CAAACATCCCCCAAAGTTCTAC-3′ 5′-TGAGGGACTTCCTCTAACCACT-3′
TBP1	forward reverse	5′-TCAAACCCAGAATTGTTCTCC-3′ 5′-AACTATGTGGTCTTCCTGAATCC-3′

## Data Availability

All relevant data are contained within the manuscript, supporting information files, and the Gene Expression Omnibus (GEO) Repository. Microarray data were deposited in GEO, accession number: GSE80223. Data are expressed as means ± SE. Statistical differences between groups were determined by unpaired *t*-test (GraphPad software) and one-way repeated measures analysis of variance (ANOVA). In all tests, differences were considered statistically significant at the 0.05 level of confidence.

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
