# Peer review of "Reciprocal Homer1a and Homer2 Isoform Expression Is a Key Mechanism for Muscle Soleus Atrophy in Spaceflown Mice"

_ijms, 2021, doi:10.3390/ijms23010075_

Round 1

Reviewer 1 Report

The current manuscript by Blottner et al. extends on the research group's previous work examining the role of Homer isoforms in response to skeletal muscle disuse. In the present study, researchers evaluated soleus and EDL muscles from rodent models following 30-days of microgravity and one-, four-, seven-, fifteen-, and twenty-one-days of ground-based hindlimb unloading. Blottner and colleagues showed that the expression of Homer isoforms is highly responsive to skeletal muscle inactivity induced by microgravity and simulated microgravity. In addition, the authors provide evidence that the localization of Homer proteins at the neuromuscular junctions (NMJs) and z-discs are affected by skeletal muscle inactivity. Altogether, the authors provide further evidence implicating Homer1a and Homer2 in the muscle remodeling process.

While the information presented in the manuscript offers additional evidence for the involvement of Homer proteins during muscle remodeling in response to skeletal muscle disuse, there are several concerns regarding the current state of the manuscript. Please see the comments below.

Concerns:

  1. The study presents data from a cohort of adult male mice (C57BL/N6; 19-20 weeks of age) flown in space for 30-days (followed by 24-48 hours of gravitational loading). The authors then present data from a set of hindlimb unloading experiments using adolescent female rats (Wistar; 6-weeks of age) unloaded for either 1, 4, 7, or 15-days. Lastly, the authors present data from a set of hindlimb unloading experiments using adolescent female rats (Sprague Dawley; 8-weeks of age) that were unloaded for 21-days. Oddly, the authors provided no explanation as to why the specific ages, genders, and strains of rats were used. Furthermore, no reason was provided as to why the particular hindlimb unloading time points (1, 4, 7, 15, and 21 days) were chosen for this study. It also seems peculiar that the authors did not include a cohort of rats unloaded for 30-days, a timepoint that could have provided a direct comparison to the space-flown mice. These choices create a lack of cohesion in the manuscript, and the fact that the authors simply glossed over these factors is very concerning.
  2. Were the expression levels for Homer 1a in Figure 1A expected to be so low? The expression results are much lower than those shown in Figure 5A, especially when comparing Figure1A FC and BG to Figure 5A CON.
  3. Accurate examination of the neuromuscular junction (NMJ) structure, innervation, and localization of associated proteins often require analysis of longitudinally oriented muscle fibers. As the authors are likely aware, this can be accomplished by either mounting the whole muscle, mounting fiber bundles, or cryosectioning muscle tissue frozen in the longitudinal orientation. In most cases, the authors only evaluated NMJs from transverse cross-sections (except Figure 7B and the upper right panel in Figure 6C). Moreover, when possible, en face NMJs should be used for quantification and presentation purposes. En face NMJs should have been used for Figure 4 so that the entire area of the postsynapse could be evaluated, rather than the use of 8 micrometer transverse cryosections that only capture a portion of the postsynapse. The following chapter is an example of one of a few different resources that provide instruction for tissue preparation, staining, and analysis of the NMJ: Pratt et al., Methods Mol Biol., 2018, PMID: 29067656.
  4. Figure 3 should be presented along with Figure 2.
  5. Section 2.5: The authors chose to cite fiber typing data and cross-sectional area data from a mouse study to support their data from rats. The soleus muscle from rats and mice differs in fiber composition, and the extent of atrophy will likely vary from a rat unloaded for 15-days versus a mouse that was flown in space for 30-days. Please provide the cross-sectional area and fiber type data from the rats used in the current study.
  6. Figure 5: HU d15 shows a dramatic decrease in expression of all Homer isoforms. The authors do not provide an explanation or even speculation about this issue. Has the muscle established a new baseline by the 15th day of unloading? Will expression continue to decrease? How does this data relate to that of the space-flown mice presented in Figure 1?
  7. Homer isoforms appear to be early responders to changes in loading, as shown by the current data. Do the authors think that the isoforms react just as quickly to reloading following a period of unloading? Notably, the space-flown mice experienced microgravity for 30-days followed by 24-48 hours of re-loading. How might this re-loading have influenced the quickly responding Homer isoforms?
  8. The antibodies used in this study detected bands with an apparent molecular weight of 45-48 kDa. And the authors suggest that the 43-45 kDa molecular weight bands refer to the longer isoforms. How did the authors confirm that the 43-45 kDa molecular weight bands refer to long Homer isoforms? In their 2011 FASEB paper, the pan Homer antibody detected a band at 48 kDa, while the Homer 2a/b antibody also detected a 48 kDa band (Salanova et al., FASEB J., 2011, PMID:21885651). Homer 1b/c and Homer 2a/b have also been seen at 48 kDa (Szumlinski et al., Neuropsychopharmacology, 2008, PMID: 17568396) and 47 kDa (Gould et al., Addict Biol., 2015, PMID: 24118426). Soleus muscles from Homer 1 knockout (KO) mice displayed Homer bands at 45 and 47 kDa (Stiber et al., Mol Cell Biol., 2008, PMID: 18268005). Of note, the same study found that Homer protein was not detected in the predominately fast-twitch plantaris from Homer 1 KO mice (Stiber et al., Mol Cell Biol., 2008, PMID: 18268005). The authors should perform western blots on soleus and EDL muscle samples from their Homer 2 KO mice. This data, along with that from the Homer 1 KO study, would help provide clarity related to the molecular weight of the various Homer isoforms. Moreover, this data could bolster the authors' claims about the molecular weight of the longer isoforms.
  9. Figure 6C – The authors state in the figure caption that “the z-disc (cross-striated pattern) was less evident in 3-weeks HU compared to CON.” Upon close examination of your images in figure 6C, it appears that you chose to show transverse cross-sections from the HU muscles rather than a longitudinal section like that shown for the CON in the upper left. It is challenging to discern a cross-striated pattern from any transverse muscle section. Please present longitudinally oriented sections co-stained with a z-disc marker such as alpha-actinin and the Homer antibody.
  10. Please provide the number of animals or samples used for each type of analysis.
  11. Throughout the manuscript, the authors discuss data that is not shown. Please show all of the data associated with the study. Including the Flown Control (FC) data in Figures 2 and 3, EDL data for Figures 1, 5, 6, 7, and fluorescent images of sections stained with only secondary antibodies.

Author Response

Authors’ response to reviewer comments

Dear Editor in Chief Prof. Prof. Dr. Maurizio Battino,

Dear Guest Editors Prof. Dr. John Lawler, Dr. Khaled Kamal

Thank you very much for the critical review and for encouraging us to pursue an accurate revision of the manuscript based on a series of constructive reviewers’ criticism. We have taken into considerations all reviewers’ comments and concerns and we have tried to respond on a point-by-point basis to many of them. Overall, we feel confident that we could provide most of the requested answers and explanations. This resulted in an improved presentation of our work, making it, hopefully, more suitable for publication in the “International Journal of Molecular Sciences” (IJMS).

Author response to Reviewer #1

Author response: Thank you very much for the critical and constructive review of our work.

Reviewer comment # 1: The study presents data from a cohort of adult male mice (C57BL/N6; 19-20 weeks of age) flown in space for 30-days (followed by 24-48 hours of gravitational loading). The authors then present data from a set of hindlimb unloading experiments using adolescent female rats (Wistar; 6-weeks of age) unloaded for either 1, 4, 7, or 15-days. Lastly, the authors present data from a set of hindlimb unloading experiments using adolescent female rats (Sprague Dawley; 8-weeks of age) that were unloaded for 21-days. Oddly, the authors provided no explanation as to why the specific ages, genders, and strains of rats were used. Furthermore, no reason was provided as to why the particular hindlimb unloading time points (1, 4, 7, 15, and 21 days) were chosen for this study. It also seems peculiar that the authors did not include a cohort of rats unloaded for 30-days, a time point that could have provided a direct comparison to the space-flown mice. These choices create a lack of cohesion in the manuscript, and the fact that the authors simply glossed over these factors is very concerning.

Author response concern 1: Thank you to the reviewer for the important observation. We fully agree with the reviewer concern 1 on our experimental design by using two different species, strains and gender at different time intervals/age in each experimental group. Unfortunately, the use of mice for hindlimb unloading (HU) experiments has been limited by the fact that in EU labs the hindlimb unloading experimental model is not allowed for more than a maximum of two weeks utilizing mostly rats rather than mice. Indeed, mice were rarely used in HU experiments. As far as we know, up to date hindlimb unloading with mice was ethically approved only for studies aimed to antagonize severe psychic depression.

Moreover, our unpublished evidence shows that female and male rats develop comparable levels of muscle atrophy after 7 days of HU. In particular, female rats are more widely used than males one, because of their higher tolerance of the experimental conditions and the lighter body weight. This latter feature has the advantage to decrease the occurrence of tail wounds, an obligatory life end-point, and, therefore, to appropriately refine the number of animals enrolled in the experimental plan.

With regard to the age of the animals, in the case of mice and laboratory rats, adulthood is deemed to be attained when sexual maturity occurs. Thus, mice are able to reproduce 52 days after birth, whereas rats only 45 days after birth. We point out that our HU rats were 3 months old.

The analysis of early unloading time-points is crucial in ruling out secondary, atrophy-driven effects on microgravity-induced changes of homer isoform expression.

Notably, Homer genes are highly conserved between mouse, rat and human. Since the main objective of this study was to address whether or not and to what extent or magnitude the expression of Homer proteins is changed following chronic disuse on Earth (simulated microgravity) vs spaceflight (real microgravity), differences were always compared between control and the same treated animal groups, strain, gender and age.

Reviewer comment # 2: Were the expression levels for Homer 1a in Figure 1A expected to be so low? The expression results are much lower than those shown in Figure 5A, especially when comparing Figure1A FC and BG to Figure 5A CON.

Author response to concern 2: In Figure 1A, data are expressed as fold changes to housekeeping reference gene PPIA whereas in Figure 5A, after normalization to PPIA, data were normalized to each control, which was set to 1. This was needed in order to compare all genes in the same plot. Otherwise, 3 different plots would have been needed to give the very different expression levels of the 3 different Homer genes.

Thus, basically, comparable levels of Homer 1a transcripts are expressed in SOL muscle of both mice and rats animals.

Reviewer comment # 3: Accurate examination of the neuromuscular junction (NMJ) structure, innervation, and localization of associated proteins often require analysis of longitudinally oriented muscle fibers. As the authors are likely aware, this can be accomplished by either mounting the whole muscle, mounting fiber bundles, or cryosectioning muscle tissue frozen in the longitudinal orientation. In most cases, the authors only evaluated NMJs from transverse cross-sections (except Figure 7B and the upper right panel in Figure 6C). Moreover, when possible, en face NMJs should be used for quantification and presentation purposes. En face NMJs should have been used for Figure 4 so that the entire area of the postsynapse could be evaluated, rather than the use of 8 micrometer transverse cryosections that only capture a portion of the postsynapse. The following chapter is an example of one of a few different resources that provide instruction for tissue preparation, staining, and analysis of the NMJ: Pratt et al., Methods Mol Biol., 2018, PMID: 29067656.

Author response to concern 3: We fully agree with the reviewer that the use of longitudinal plane of NMJ would have been optimal for our Homer protein subcellular localization analysis. However, due to the scarcity of muscle material from spaceflown mice, we were not able to prepare, stain and study NMJ as necessary for standard laboratory animals. A sentence was added to page 5, lines 158-161.

On these foregoings, a more accurate Homer protein NMJ subcellular localization and 3D-structure analysis on 30µm cryosections were done in one of our recent works (Lorenzon P. et al., Metabolites 2021).

In our previous work, triplicate 8µm cryosections were sufficient to detect changes in Homer fluorescence intensities at the NMJ postsynaptic microdomains (Salanova et al, FASEB J. 2011).

Despite this, we have cited the reference of Pratt et al. for awareness of NMJ analyses.

Reviewer comment # 4: Figure 3 should be presented along with Figure 2.

Author response to concern 4: As suggested by the reviewer, in the revised version of the manuscript Figure 3 is presented together with Figure 2.

Reviewer comment # 5: Section 2.5: The authors chose to cite fiber typing data and cross-sectional area data from a mouse study to support their data from rats. The soleus muscle from rats and mice differs in fiber composition, and the extent of atrophy will likely vary from a rat unloaded for 15-days versus a mouse that was flown in space for 30-days. Please provide the cross-sectional area and fiber type data from the rats used in the current study.

Author response concern 5: We already provided data concerning MW/BW ratio for Wistar rats unloaded for 1-15 days in Figure 5B. These data show that muscle mass decreases significantly from HU day 4 (ANOVA P<0.001). A further significant reduction in soleus mass was observed between HU7 and HU15 groups (P=0.04). These data were added to the Figure legend. We then measured fiber cross-sectional area to validate the lack of mass loss observed after 1d HU. The analysis showed that HU 1d does not induce a significant reduction of cross-sectional area in both slow and fast fibers, confirming our previous finding (Lechado i Terradas et al. 2018).

Unloading-induced changes in fiber-type composition were investigated in ambulatory and HU7 and HU15 groups. Results show the lack of significant difference in the percentage of fast fibers, which correspond to about 10-20% of total fibers. Such a relatively high level is compatible with the young age of the animals under investigation.

Measurements also took into account the population of intermediate fibers, expressing both slow and fast myosin. Consistently with literature data, a significantly higher percentage of intermediate fibers (index of fiber remodeling / transition / shift) was detected in HU15 rat group.

Reviewer comment # 6: Figure 5: HU d15 shows a dramatic decrease in expression of all Homer isoforms. The authors do not provide an explanation or even speculation about this issue. Has the muscle established a new baseline by the 15th day of unloading? Will expression continue to decrease? How does this data relate to that of the space-flown mice presented in Figure 1?

Author response to concern 6: We fully agree with reviewer concern 6 that there are substantial difference between HU at Earth gravity and microgravity exposure in Space and that the relative effects are difficult to compare. However, both are well-accepted experimental models of disuse-induced muscle atrophy.

Previous studies on HU experimental models showed that Homer expression, especially in skeletal muscle soleus, is linked to the ubiquitine-3-ligase proteasome pathway (Bortoloso E. et al., Am J Physiol Cell Physiol 304:C68-C77, 2013). Consistent with this, as inferred from global gene array analysis, Homer genes expression changes were found in SOL muscle but not in EDL from the same animal groups (Gambara et al., PLoS One. 2017 Jan 11;12(1):e0169314).

What really happens at the Homer molecular level after two weeks of HU, it cannot be said with any certainty since our mRNA analysis was carried out up to 15 days of HU. Homer protein expression was investigated at 3 weeks HU (Figure 5 revised manuscript).

We assume that after two weeks of HU there is a progressive reduction of muscle atrophy. However, the molecular mechanisms responsible are not yet identified.

Reviewer comment # 7: Homer isoforms appear to be early responders to changes in loading, as shown by the current data. Do the authors think that the isoforms react just as quickly to reloading following a period of unloading? Notably, the space-flown mice experienced microgravity for 30-days followed by 24-48 hours of re-loading. How might this re-loading have influenced the quickly responding Homer isoforms?

Author response concern 7: Thank you to the reviewer for this important issue. We are fully agree with the reviewer`s criticism and we cannot fully exclude the hypothesis that observed changes for Homer1a in space-flown mice stem from muscle re-loading since we haven`t done control experiments by using hindlimb unloading rat muscles followed by 24-48 hours re-loading.

However, assuming that this were the case, we would have expected the long forms of Homer proteins to be increased rather than the dominant negative inducible Homer1a isoform.

Moreover, after 30 days of microgravity exposure, we do not think that a full re-loading is attained just after landing and within hours post landing due to observed animal behavioral constraints, e.g., marked hypokinesia with few limb extensions against gravity.

Reviewer comment # 8: The antibodies used in this study detected bands with an apparent molecular weight of 45-48 kDa. And the authors suggest that the 43-45 kDa molecular weight bands refer to the longer isoforms. How did the authors confirm that the 43-45 kDa molecular weight bands refer to long Homer isoforms? In their 2011 FASEB paper, the pan Homer antibody detected a band at 48 kDa, while the Homer 2a/b antibody also detected a 48 kDa band (Salanova et al., FASEB J., 2011, PMID:21885651). Homer 1b/c and Homer 2a/b have also been seen at 48 kDa (Szumlinski et al., Neuropsychopharmacology, 2008, PMID: 17568396) and 47 kDa (Gould et al., Addict Biol., 2015, PMID: 24118426). Soleus muscles from Homer 1 knockout (KO) mice displayed Homer bands at 45 and 47 kDa (Stiber et al., Mol Cell Biol., 2008, PMID: 18268005). Of note, the same study found that Homer protein was not detected in the predominately fast-twitch plantaris from Homer 1 KO mice (Stiber et al., Mol Cell Biol., 2008, PMID: 18268005). The authors should perform western blots on soleus and EDL muscle samples from their Homer 2 KO mice. This data, along with that from the Homer 1 KO study, would help provide clarity related to the molecular weight of the various Homer isoforms. Moreover, this data could bolster the authors' claims about the molecular weight of the longer isoforms.

Author response concern 8: We apologise for the mistake. A closer look at Figure 6a shows that the molecular weights pattern of Homer proteins is compatible with 45-48 kDa rather than 43-45 kDa, which fully overlaps with the MWs reported in our previous work and that of other authors.

Reviewer comment # 9: Figure 6C – The author’s state in the figure caption that “the z-disc (cross-striated pattern) was less evident in 3-weeks HU compared to CON.” Upon close examination of your images in figure 6C, it appears that you chose to show transverse cross-sections from the HU muscles rather than a longitudinal section like that shown for the CON in the upper left. It is challenging to discern a cross-striated pattern from any transverse muscle section. Please present longitudinally oriented sections co-stained with a z-disc marker such as alpha-actinin and the Homer antibody.

Author response concern 9: We fully agree with the reviewer comments 9. Overall, this sentence is referred to all results obtained of the HU composite analysis and the selection of the images was taking into account the orientation of the NMJ AChRs distribution rather than the orientation of the myofiber. Therefore, considering that the present work focused mainly on the neuromuscular junction, and we have not showing any evidence at the the Z-disc level, which is, basically, the subject of another study currently ongoing, we have removed the sentence referring to the Z-disc (page 10, line 247).

Moreover, clear evidence demonstrating that Homer proteins are subcellularly localized at the Z-disc of mouse muscle has already been shown in Figure 1G (Homer/desmin co-localization) of one of our previous studies (Salanova M. et al., Cell Calcium 2002).

Reviewer comment # 10: Please provide the number of animals or samples used for each type of analysis.

Author response to concern 10: As suggested by the reviewer, the number of animals used for the analysis reported in Material and Methods section was inserted in the figure legend.

Reviewer comment # 11: Throughout the manuscript, the authors discuss data that is not shown. Please show all of the data associated with the study. Including the Flown Control (FC) data in Figures 2 and 3, EDL data for Figures 1, 5, 6, 7, and fluorescent images of sections stained with only secondary antibodies.

Author response to concern 11: Data from both SOL and EDL muscles from BF, BG and FC are included in Figures 1, 3 and 4. Since the real reference control for Bion-M1 flight (BF) is the Bion-M1 animal facility on Earth gravity (BG) rather than the standard cage control (FC) on ground, thus, a direct comparison of BF with BG is provided at the histochemical level in Figure 2.

Internal negative controls images stained with only secondary antibodies are provided below.

All changes reported in the manuscript are highlighted in red by the Microsoft tracking changes!

Reviewer 2 Report

Blottner et al provide a nice follow-up study of their previous work on Homer2 regulation during conditions promoting muscle disuse atrophy. The authors show a downregulation of Homer2 protein accumulation at the NMJ in a cohort of mice subjected to microgravity, and compare this to mice and rats subjected to hindlimb unloading. Time course analysis in the latter model suggests Homer2 downregulation preceding muscle atrophy, which is expected given the role of Homer2 antagonizing muscle protein degradation (Bortoloso et al. 2012). The study adds to previous studies reporting overlapping findings in other models of disuse atrophy. Given this, however, novelty of the study is limited.

The study is technically well conducted and clearly written, and I only have few remarks to improve data presentation.

  1. In general, N-numbers for animals used should be provided in the figure legends.
  2. Statistical testing: Fig. 1, 3, 5A: would not ANOVA be better for this type of data?
  3. General: instead of “dynamite plunger plots” I would strongly encourage plotting individual data points.

Author Response

Authors’ response to reviewer comments

Dear Editor in Chief Prof. Prof. Dr. Maurizio Battino,

Dear Guest Editors Prof. Dr. John Lawler, Dr. Khaled Kamal

Thank you very much for the critical review and for encouraging us to pursue an accurate revision of the manuscript based on a series of constructive reviewers’ criticism. We have taken into considerations all reviewers’ comments and concerns and we have tried to respond on a point-by-point basis to many of them. Overall, we feel confident that we could provide most of the requested answers and explanations. This resulted in an improved presentation of our work, making it, hopefully, more suitable for publication in the “International Journal of Molecular Sciences” (IJMS).

Author responce Reviewer #2

Author response: Thank you very much for the critical and constructive review of our work.

Reviewer comment # 1 In general, N-numbers for animals used should be provided in the figure legends.

Author response to concern 1: As suggested from the reviewer, the number of animals used for the analysis reported in Material and Methods section was inserted in the figure legend.

Reviewer comment # 2: Statistical testing: Fig. 1, 3, 5A: would not ANOVA be better for this type of data?

Author response to concern 2: Statistic ANOVA test analysis was add to the Figure 4 of the revised version of the Manuscript

Reviewer comment # 3: General: instead of “dynamite plunger plots” I would strongly encourage plotting individual data points.

Author response to concern 3: As suggested by the reviewer, original graphs for figure 1 and Figure 3 have been replaced by graphs plotting individual data points.

All changes reported in the manuscript are highlighted in red by the Microsoft tracking changes!

Round 2

Reviewer 1 Report

After reviewing the author's responses and the revised manuscript, several issues remain.

  1. Justify the use of N = 2 and N = 3 for the mouse experiments (Revised Figures 1, 2, & 3). Provide sample size calculations.
  2. Be consistent with data presentation.

    1. All graphs should be similar to Revised Figures 1 & 3, which display individual data points. This is especially important when presenting findings from such small groups (n = 2, n = 3, n = 4).

    2. Revised Figure 1 and Revised Figure 5A should be presented in the same way. For instance, normalize to PPIA and then normalize to control.

    3. Revised Figure 2, Panel A – Show the FC images. Data from the FC group is presented in Revised Figures 1, 2B, and 3. There is no reason to leave out the FC data in Figure 2A.

  3. Were single 8 μm cryosections or three serial 8 μm cryosections used for NMJ quantification? After reading the Author Response to concern 3 and reviewing citation #16, the 2011 FASEB paper does not appear to state that three serial 8 μm cryosections were used for quantification in that paper. Furthermore, the authors do not state in the current manuscript that three serial 8 μm cryosections were used for NMJ quantification.

  4. Revised Figure 3 and Revised Figure 2 – were statistics performed using the pixel intensity values from each NMJ or the average of the NMJ per mouse (n = 2)? It is more appropriate to analyze and present the value from each mouse or the average value from each mouse rather than displaying each individual value from NMJs within a mouse. The same approach applies to skeletal muscle fibers and cross-sectional area analysis.

  5. Muscle weight / body weight and skeletal muscle fiber cross-sectional area (CSA) are not equivalent measurements. Skeletal muscle tissue is comprised of more than muscle tissue. Several other cell types are present, including fat, neurons, inflammatory cells, fibroblasts, vasculature, to list a few. The proper way to characterize atrophy is to assess changes in CSA of individual muscle fibers, preferably demonstrating fiber type. The MW/BW data shows no reduction at HU 1D, and, unsurprisingly, the Revised CSA data shows no difference between HU 1D and controls. In the revised manuscript (Lines 226-228), the authors compare the HU rat MW/BW to decreased CSA by fiber type from 30-days of space flight in mice. Provide the muscle fiber CSA from the HU4, HU7, and HU15 time points for a more appropriate comparison.

  6. In response to "Author response concern 1" – Please provide a citation or website link for the limitations imposed on EU labs concerning rodent hindlimb unloading experiments.

  7. In response to "Author response concern 1" – Half of the unloading studies were performed in China. Were the EU hindlimb unloading experimental limitations also imposed on these studies?
  8. In response to "Author response concern 1" – The authors state, “We point out that our HU rats were 3 months old.” Please review Section 5.3 Hind limb unloading (HU) experimental animals and animal care in the revised manuscript. Line 480 – “six-week-old female Wistar rats.” Line 490 – “eight-week old female Sprague Dawley.”

  9. In response to "Author response concern 1" – Anecdotal evidence is not acceptable. Provide citations for the following:

    1. Atrophy levels between male mice and female rats.

    2. Atrophy levels between Wistar rats and Sprague Dawley rats.

    3. Age of adulthood in mice and rats.

Author Response

Authors’ response to reviewer comments

Author responce Reviewer #1

Author response: Thank you very much for the very critical and constructive review to our work.

Reviewer comment # 1: Justify the use of N = 2 and N = 3 for the mouse experiments (Revised Figures 1, 2, & 3). Provide sample size calculations.

Author response concern 1: As part of the international tissue sharing group program, all together, we had leg muscles from 5 different mice that were housed aboard the Bion-M1 facility for 30 days on orbit.

Since one of our main objectives was to investigate in animal exposed to microgravity at global gene expression level we dedicated muscles from 3 space-flown mice for “Global Gene Array” analysis and muscles from 2 mice for morphological and structural analysis. Part of the result of these experiments including myofiber CSA, are reported in the following publication of ours:

-Gambara G, Salanova M, Ciciliot S, Furlan S, Gutsmann M, Schiffl G, Ungethuem U, Volpe P, Gunga HC, Blottner D. PLoS One. 2017 Jan 11;12(1):e0169314. doi: 10.1371/journal.pone.0169314..

Reviewer comment # 2: Be consistent with data presentation.

  1. All graphs should be similar to Revised Figures 1 & 3, which display individual data points. This is especially important when presenting findings from such small groups (n = 2, n = 3, n = 4).
  2. Revised Figure 1 and Revised Figure 5A should be presented in the same way. For instance, normalize to PPIA and then normalize to control.
  3. Revised Figure 2, Panel A – Show the FC images. Data from the FC group is presented in Revised Figures 1, 2B, and 3. There is no reason to leave out the FC data in Figure 2A.                                          

Author response concern 2: As suggested by the reviewer:                                                               a. All graphs have been changed as scatter plot.

  1. We assumed the Reviewer is referring to Figure 4A and not 5A. Revised Figure 1 and Revised Figure 4A are now presented in the same way.
  2. Confocal images from the FC group have been included in Figure 2A.

Reviewer comment # 3: Were single 8 μm cryosections or three serial 8 μm cryosections used for NMJ quantification? After reading the Author Response to concern 3 and reviewing citation #16, the 2011 FASEB paper does not appear to state that three serial 8 μm cryosections were used for quantification in that paper. Furthermore, the authors do not state in the current manuscript that three serial 8 μm cryosections were used for NMJ quantification.

Author response concern 3: In the first round of the revision (Author response to concern 3), we wrote: “In our previous work, triplicate 8 µm cryosections were sufficient to detect changes in Homer fluorescence intensities at the NMJ postsynaptic microdomains (Salanova et al, FASEB J. 2011)” intending that data from single 8 µm cryosections were sufficient to detect changes in Homer fluorescence intensities at the NMJ postsynaptic microdomains and that data were collected from at least 3 independent experiments.  

As in Salanova et al, FASEB J. 2011, in the present work, single 8 µm cryosections were analysed. To make this point clearer, we revised the text (line 512-517).

Reviewer comment # 4: Revised Figure 3 and Revised Figure 2 – were statistics performed using the pixel intensity values from each NMJ or the average of the NMJ per mouse (n = 2)? It is more appropriate to analyze and present the value from each mouse or the average value from each mouse rather than displaying each individual value from NMJs within a mouse. The same approach applies to skeletal muscle fibers and cross-sectional area analysis.

Author response concern 4: Data are expressed as mean from pooled data.

Reviewer comment # 5: Muscle weight / body weight and skeletal muscle fiber cross-sectional area (CSA) are not equivalent measurements. Skeletal muscle tissue is comprised of more than muscle tissue. Several other cell types are present, including fat, neurons, inflammatory cells, fibroblasts, vasculature, to list a few. The proper way to characterize atrophy is to assess changes in CSA of individual muscle fibers, preferably demonstrating fiber type. The MW/BW data shows no reduction at HU 1D, and, unsurprisingly, the Revised CSA data shows no difference between HU 1D and controls. In the revised manuscript (Lines 226-228), the authors compare the HU rat MW/BW to decreased CSA by fiber type from 30-days of space flight in mice. Provide the muscle fiber CSA from the HU4, HU7, and HU15 time points for a more appropriate comparison.

Author response concern 5: We agree that MW/BW ratio is not equivalent to myofiber CSA measurements. The second one is a much more sensitive index. This is the reason why we provided CSA measurement after 1-day hindlimb unloading, in order to exclude the presence of myofiber atrophy at this early time point. As reported in a previous publication (Lechado iTerradas et al.2018, ref. 34), CSA may decrease (as it occurs in soleus muscle after 2-day unloading) even in the absence of a statistically significant reduction of MW/BW ratio. Conversely, a significantly decreased MW/BW ratio occurred concomitantly with decreased CSA values, as we observed after 7 day unloading, when soleus type-I and –IIA fibers showed a  -46% and -38% reduction, respectively (Vitadello et al. J Physiol, 592: 2637-2652, 2014). Therefore, the comparison with CSA data concerning space flown mice is substantially correct, since a significantly reduced MW/BW ratio indeed represents a reliable index of muscle atrophy. 

The sentence at line 224 was rewritten as follows:                                                                 “Compared to CON animals, normalized SOL muscle mass to body weight (MW/BW) appeared reduced by about 25% after HU 4- 7days and by 40% after the second week of HU (ANOVA p<0.001, Figure 4B). The CSA (cross-sectional area) analysis confirmed the absence of significant muscle atrophy after HU1d (Figure 2S), as already published for another animal group (34). In a  previous study we reported that a -36% reduction of MW/BW ratio was accompanied by a -46% decrease in CSA for type -I myofibers and -38% for type-IIA ones (35. Vitadello, M., Germinario, E., Ravara, B., Dalla Libera, L., Danieli-Betto, D., & Gorza, L. (2014). Curcumin counteracts loss of force and atrophy of hindlimb unloaded rat soleus by hampering neuronal nitric oxide synthase untethering from sarcolemma. The Journal of Physiology592, 2637–2652.). This reduction…..”

Reviewer comment # 6: In response to "Author response concern 1" – Please provide a citation or website link for the limitations imposed on EU labs concerning rodent hindlimb unloading experiments.

Author response concern 6:

As reported in section 4.3, lines 477 to 486, we confirm that: the two weeks HU protocol (17) was performed according to the recommendations provided by the European Convention for the Protection of Vertebrate Animals used for Experimental and Scientific Purposes (Council of Europe No. 123, Strasbourg, 1985) and authorized by the Animal Ethics 458 Committee of the University of Padua and the Italian Health Ministry (103/2007B).

To accomplish the reviewer`s request, we provide in attachment:

  1. A copy of the original document with the rejection of mice HU from the Italian Health Ministry is provided (“Health Ministry letter135”).

  1. A copy of the reply to Health Ministry letter (“Reply to Health Ministry letter136”)

Reviewer comment # 7:

In response to "Author response concern 1" – Half of the unloading studies were performed in China. Were the EU hindlimb unloading experimental limitations also imposed on these studies?

Author response concern 7:

No, they are two independent studies. See Sun LW et al., JMNI (2013); ref. 36.

Reviewer comment # 8: In response to "Author response concern 1" – The authors state, “We point out that our HU rats were 3 months old.” Please review Section 5.3 Hind limb unloading (HU) experimental animals and animal care in the revised manuscript. Line 480 – “six-week-old female Wistar rats.” Line 490 – “eight-week old female Sprague Dawley.”

Author response concern 8:

We apologize for the mistake! As reported in section 4.3 in the present study we used six-week-old female Wistar and eight-week-old female Sprague Dawley!

Reviewer comment # 9: In response to "Author response concern 1" – Anecdotal evidence is not acceptable. Provide citations for the following:

  1. Atrophy levels between male mice and female rats.
  2. Atrophy levels between Wistar rats and Sprague Dawley rats.
  3. Age of adulthood in mice and rats.

Author response concern 9:

  1. Atrophy levels between male mice and female rats.

A literature survey performed before to start out experimental setting of hindlimb unloading (around 2002-2003) showed that the use of female rats appeared more widespread than the use of male rats.  Nevertheless, 7 day-hindlimb unloading induces a -27% and -28% reduction in soleus MW/BW ratio of  Wistar male and female rats, respectively, as illustrated in the enclosed box plot for male muscles and consistently with data shown in Figure 4B for females ones.   

  1. Atrophy levels between Wistar rats and Sprague Dawley rats?                             
    1. CSA two-week-animal unloading: please see “Author response concern 5”

  1. CSA three-week-animal unloading: no comparative studies were performed to assess the presence of difference in muscle atrophy degree between these two rat strains, since a detailed myofiber CSA analysis of three-week HU carried out in a previous study (Sun et al., J Musculoskelet Neuronal Interact 2013; 13(2):166-177) showed that fiber atrophy (decreased CSA) of type I and type II myofibers was found in 21d HU rat solei.

The decreased myofiber cross-sectional area of the BF SOL muscle, which was -53% for type-I, -44% for type-IIA, -50% for type-IIB and -9% for type-IIX (Ref. 31).

This information was added to the manuscript, following the revision to question 5 at line 227.

  1. Age of adulthood in mice and rats.

In our animal facility, rat litters are separated after 5 weeks after birth, since some female can be capable for breeding. Reproductive ability in animals represents an adulthood feature; therefore, we consider our female rats as young adult ones.
